# Second-order spectral lineshapes from charged interfaces

Paul E. Ohno [1], Hong-fei Wang[2] & Franz M. Geiger[1]

Second-order nonlinear spectroscopy has proven to be a powerful tool in elucidating key chemical and structural characteristics at a variety of interfaces. However, the presence of interfacial potentials may lead to complications regarding the interpretation of second harmonic and vibrational sum frequency generation responses from charged interfaces due to mixing of absorptive and dispersive contributions. Here, we examine by means of mathematical modeling how this interaction influences second-order spectral lineshapes. We discuss our findings in the context of reported nonlinear optical spectra obtained from charged water/air and solid/liquid interfaces and demonstrate the importance of accounting for the interfacial potential-dependent $\chi^{(3)}$ term in interpreting lineshapes when seeking molecular information from charged interfaces using second-order spectroscopy.

[1] Department of Chemistry, Northwestern University, Evanston, IL 60208, USA. [2] Department of Chemistry and Shanghai Key Laboratory of Molecular Catalysis and Innovative Materials, Fudan University, Shanghai 200433, China. Correspondence and requests for materials should be addressed to H.-f.W. (email: wanghongfei@fudan.edu.cn) or to F.M.G. (email: geigerf@chem.northwestern.edu)

Following early work considering phase relationships in second harmonic generation (SHG) responses from charged aqueous interfaces[1], a formalism was derived recently[2, 3] and subsequently established experimentally[4] in which the presence of interfacial potentials may lead to the mixing of second- and third-order contributions to nonlinear optical responses from charged interfaces[4, 5]. Specifically, in non-resonant SHG experiments performed at the z-cut α-quartz/water interface, the interfacial potential, $\Phi(0)$, was found to interact with the second- and third- order susceptibilities of the interface, $\chi^{(2)}$ and $\chi^{(3)}$, both of which are purely real under the non-resonant conditions of the experiment, according to[4, 5]

$$\chi_{total}^{(2)} \propto \chi^{(2)} + (\chi_1^{(3)} + i\chi_2^{(3)})\Phi(0) \quad (1)$$

As described in Supplementary Note 1, we have modified the sign in the phase matching factor in equation (1) from - to + when compared to our previous derivation[4, 5] so as to be consistent with the literature[2, 3]. Under conditions of electronic or vibrational resonance, for which $\chi^{(2)}$ and/or $\chi^{(3)}$ are not purely real anymore, the interfacial second- and third-order terms may further mix[4, 5].

Here, we provide the formalism for and examine by means of simulations how the mixing displayed in equation (1) may influence spectral lineshapes under absorptive (resonant) conditions. We derive that the $\chi^{(3)}$ phase angle $\varphi$ contains important physical parameters pertaining to the experimental geometry and the Debye screening length, thus opening a path to testing existing electrical double layer theories against an experimental measurable. Moreover, we find that mixing between the absorptive and dispersive terms that results from an interfacial potential may significantly contribute to features observed in reported potential-dependent vibrational sum frequency generation (SFG) spectra of single oscillators, multiple oscillators, and continua of oscillators, such as those relevant for aqueous interfaces. In addition, we present methods to account for absorptive-dispersive mixing in nonlinear optical responses from charged interfaces under conditions of electronic or vibrational resonance. Finally, our formalism provides a means to perform counterfactuals in atomistic simulations of the nonlinear optical responses of charged interfaces, i.e. spectra can now be computed with and without absorptive-dispersive interactions so that comparison to experiments can be made more reliably.

## Results

### Quantifying absorptive-dispersive interactions.
We simulate SFG spectra by using equation (1) with an absorptive (resonant) $\chi_{res,\nu}^{(2)}$ term that takes the following form for $N_{ads}$ oscillators having a resonant frequency $\omega_\nu$:

$$\chi_{res,\nu}^{(2)} \propto N_{ads} \left\langle \frac{A_\nu M_\nu}{\omega_{IR} - \omega_\nu + i_\nu} \right\rangle \quad (2)$$

Here, $A_\nu$ and $M_\nu$ are the Raman transition probability and the IR transition dipole moment of the oscillator, respectively, $\omega_{IR}$ is the infrared frequency of the incident probe light, $\Gamma_\nu$ is the damping factor of the mode (related to its observed bandwidth, or lifetime), and the brackets indicate averaging over all molecular orientations. Multiple oscillators are represented by a sum over $\nu = 1$ to $n$ modes that interact through their phases, $\gamma_\nu$[6-8]. Combining equation (1) with equation (2) then yields, in the presence of an interfacial potential, $\Phi(0)$, the following expression

for the SFG signal intensity:

$$I_{SFG} \propto \left| \chi_{NR}^{(2)} + \sum_{\nu=1}^{n} \chi_{res,\nu}^{(2)} e^{i\gamma_\nu} + \left(\chi_1^{(3)} + i\chi_2^{(3)}\right)\Phi(0) \right|^2 \quad (3a)$$

Given that $\chi_1^{(3)} = \frac{\kappa^2}{\kappa^2 + (\Delta k_z)^2}\chi^{(3)}$ and $\chi_2^{(3)} = \frac{\kappa\Delta k_z}{\kappa^2 + (\Delta k_z)^2}\chi^{(3)}$[4, 5], we have

$$\left(\chi_1^{(3)} + i\chi_2^{(3)}\right) = \left(\frac{\kappa^2}{\kappa^2 + (\Delta k_z)^2}\chi^{(3)} + i\frac{\kappa\Delta k_z}{\kappa^2 + (\Delta k_z)^2}\chi^{(3)}\right) = \frac{\kappa}{\kappa - i\Delta k_z}\chi^{(3)} \quad (3b)$$

Here, $\kappa$ is the inverse of the Debye screening length and $\Delta k_z$ is the inverse of the coherence length of the SHG or SFG process. Combining equation (3a) and equation (3b) and expressing the complex $\chi^{(3)}$ in terms of its magnitude and its phase angle, $\varphi$, we find:

$$I_{SFG} \propto \left| \chi_{NR}^{(2)} + \sum_{\nu=1}^{n} \chi_{res,\nu}^{(2)} e^{i\gamma_\nu} + \frac{\kappa}{\sqrt{\kappa^2 + (\Delta k_z)^2}} e^{i\varphi}\chi^{(3)}\Phi(0) \right|^2 \quad (3c)$$

Despite $\chi^{(3)}$ contributions being essentially neglected in earlier SFG studies of charged aqueous interfaces, recent work by Tian and Shen and coworkers[2] firmly established the importance of the $\chi^{(3)}$ term in SFG spectroscopy and attempted to separate the $\chi^{(2)}$ and $\chi^{(3)}$ contributions to the total SFG signal from the air/water interface of electrolyte aqueous solution. We emphasize here that the $\chi^{(3)}$ phase angle $\varphi$ is not the phase of the "complex $\Psi$" discussed in reference (2). Instead, the phase matching factor we provide is for the optical field, not the static field, given that static potentials are real. As the $\chi^{(3)}$ phase angle $\varphi$ is determined by the coefficients of the real and imaginary $\chi^{(3)}$ terms of equation (3b), it ultimately depends on the optical parameters and electrostatic conditions of the experiment.

Applying Euler's formula to equation (3b), we find that the $\chi^{(3)}$ phase angle $\varphi$ takes the form

$$\varphi = \arctan(\Delta k_z/\kappa) \quad (4)$$

Equation (4) reveals that the $\chi^{(3)}$ phase angle $\varphi$ contains valuable information, as it depends on the values of $\kappa$ and $\Delta k_z$ that are given by each specific condition of the sample, aqueous environment, pH, ionic strength, temperature, etc., and the geometry of the laser experiment. Indeed, given that the $\chi^{(3)}$ phase angle $\varphi$ is in principle measurable, as indicated below, and given that $\Delta k_z$ is known for a given experiment, equation (4) is a means for experimentally finding the Debye screening length as a function of charge density, ionic strength, and surface potential. As such, equation (4) provides an experimental benchmark for evaluating both common mean field theories as well as atomistic models of charged interfaces.

Provided that the $\chi^{(3)}$ phase angle $\varphi$ is known, all remaining phase angles in the $\chi_{res,\nu}^{(2)}$ terms of equation (3) can be correctly quantified using heterodyning or phase referencing methods. Unfortunately, a method for measuring the $\chi^{(3)}$ phase angle $\varphi$ in addition to the remaining phase angles for the absorptive terms has not yet been demonstrated, to the best of our knowledge. Therefore, we provide estimates for the $\chi^{(3)}$ phase angle $\varphi$ by calculating the wavelength-dependent inverse coherence length using typical experimental parameters for SHG and SFG experiments in reflection geometries (e.g., for the case of SFG,

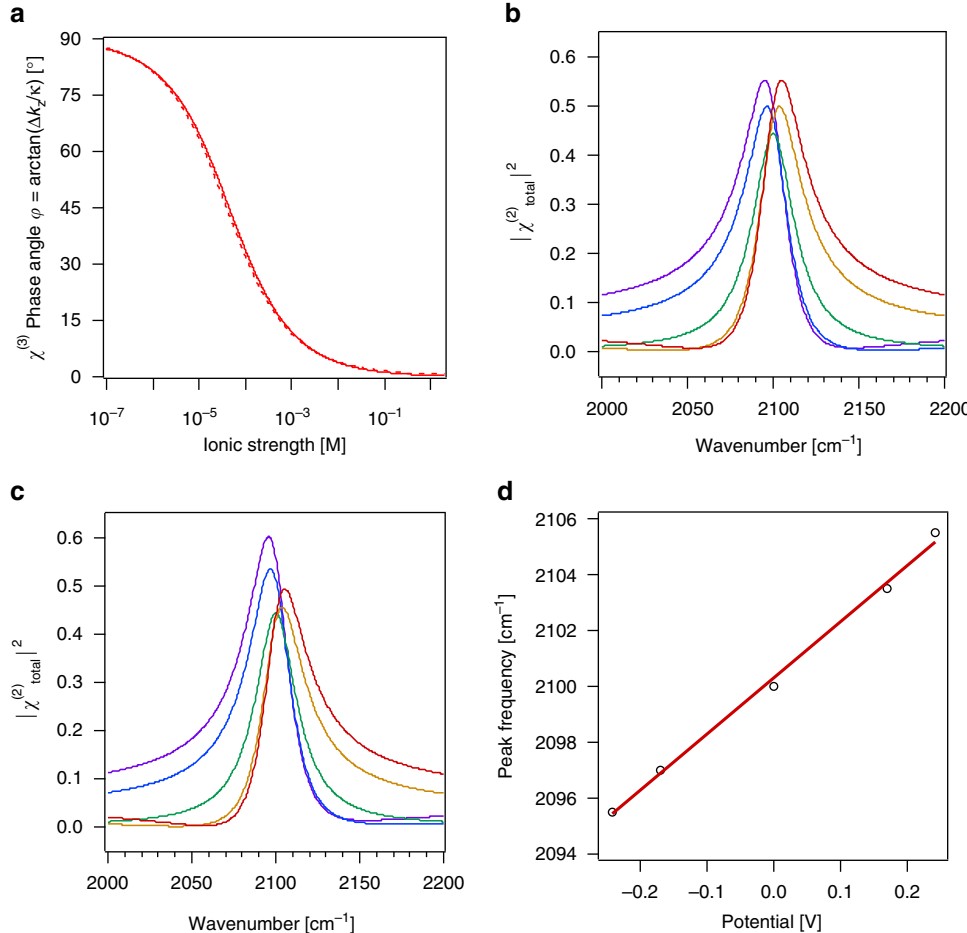

**Fig. 1** Single oscillator. **a** The $\chi^{(3)}$ phase angle $\varphi$ computed for $\omega = 3000$ cm$^{-1}$ (solid line) and 2100 cm$^{-1}$ (dashed line) from equation (4) as described in the main text for varying ionic strength. SFG intensity spectra of a single oscillator on a surface simulated for $\omega_1 = 2100$ cm$^{-1}$, $A_v M_v = 10$, $\Gamma_v = 15$ cm$^{-1}$, $\gamma_1 = 0°$, $\chi^{(3)} = 1.0$, and $\Phi(0) = -240$ mV (purple), $-170$ mV (blue), $<1$ mV (green), $+170$ mV (orange), and $+240$ mV (red), with an ionic strength of 1 mM, calculated without **b** and with **c** absorptive-dispersive interactions. **d** Peak frequency computed from equation (1) as a function of interfacial potential, $\Phi(0)$, and linear least squares fit (red line)

$\Delta k_z = \sim 1/(44 \times 10^{-9}$ m) in the OH stretching region, using incident angles of 45 degrees and 60 degrees for the upconverter and infrared frequencies, respectively, see Fig. 1a), and model the interfacial potential with Gouy-Chapman theory for a 1:1 electrolyte in aqueous solution, based on the ionic strength dependent Debye screening length, $\kappa^{-1}$. Specifically, we express the interfacial potential in the familiar form $\Phi_0 = \frac{2k_BT}{e}\sinh^{-1}\left[\frac{\sigma}{\sqrt{8000k_BTN_A\varepsilon_0\varepsilon_rC}}\right]$, where $\sigma$ is the interfacial charge density, $\varepsilon_r$ is the dielectric concentration of the solvent (for instance, 78 for bulk water), $C$ is the molar electrolyte constant, and the fundamental physical constants have their standard meanings. Given these assumptions, we find that at low ionic strength, i.e. high interfacial potentials, the $\chi^{(3)}$ phase angle approaches, but cannot be exactly, 90 degrees, as that would mean $\kappa = 0$ (infinitely long Debye length), for which $\frac{\kappa}{\sqrt{\kappa^2 + (\Delta k_z)^2}}$, and therefore the entire $\chi^{(3)}$ term, goes to zero. Water autoionization puts a limit on the lowest ionic strength in water ($1 \times 10^{-7}$ M) and therefore an upper limit for the Debye length (961 nm) and a lower limit for $\kappa$ ($1.0 \times 10^6$ m$^{-1}$), using, for instance, our Gouy-Chapman model assumptions discussed above. Therefore, the maximum value that the $\chi^{(3)}$ phase angle can assume is approximately 88 degrees. Conversely, for our example, the lowest value would be approximately 2 degrees for brine conditions.

Our estimation shows that it is worthwhile to develop an experimental method for measuring the $\chi^{(3)}$ phase angle $\varphi$ directly, as the sensitivity of equations (3) and (4) to variations in, say, the relative permittivity or the surface charge density can be large. For instance, using a lower relative permittivity (for instance, a value of 40 or 20)[9–14] as opposed to 78 in our Gouy-Chapman model example increases the magnitude of the interfacial potential and thus lowers $\kappa$, which then strongly influences the value of the $\chi^{(3)}$ phase angle $\varphi$ per equation (4).

In what follows, we hold any non-resonant $\chi^{(2)}$ response, $\chi^{(2)}_{NR}$, at zero and explore the interaction of the $\chi^{(2)}_{res,v}$, and $\chi^{(3)}$ terms for varying values of charge density and ionic strength, and thus interfacial potential, $\Phi(0)$. We begin with cases in which the $\chi^{(3)}$ contributions are purely real. These cases are to recapitulate scenarios in which a species with an electronic or vibrational resonance matching the second harmonic or the sum frequency wavelength is adsorbed, while there are no such species in the bulk phase. This might be the case of carbon monoxide or a methyl group on a surface in contact with a medium that contains no CO or CH oscillators. We then proceed to examine cases where the $\chi^{(3)}$ as well as the $\chi^{(2)}$ contributions contain resonances. These cases might well represent scenarios in which molecules with electronic or vibrational resonances at the second harmonic or the sum frequency wavelengths are present both at the surface and the bulk, such as water/solid or water/air interfaces in terms

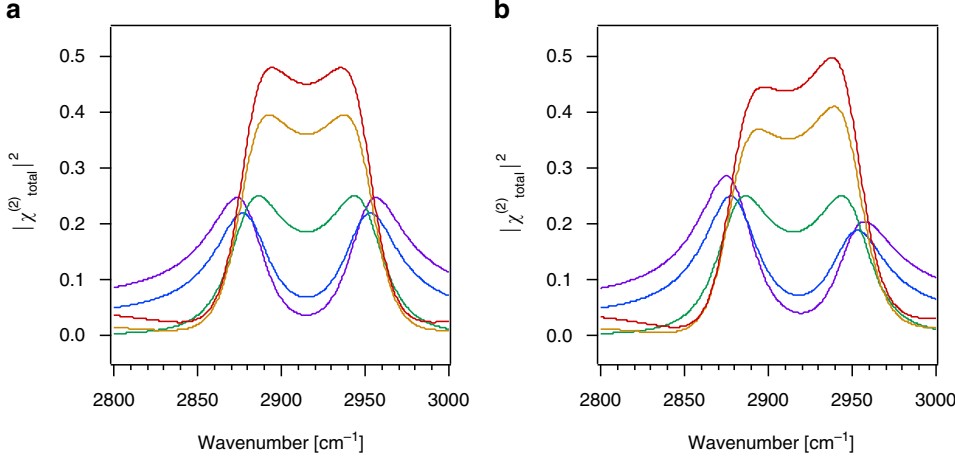

**Fig. 2** Dual oscillators. SFG intensity spectra of two oscillators on a surface simulated for $\omega_1 = 2880$ cm$^{-1}$, $\gamma_1 = 0°$, $\omega_2 = 2950$ cm$^{-1}$, $\gamma_2 = 180°$, $A_v M_v = 10$, $\Gamma_v = 20$ cm$^{-1}$, $\chi^{(3)} = 1$, and $\Phi(0) = -240$ mV (purple), $-170$ mV (blue), $< 1$ mV (green), $+170$ mV (orange), and $+240$ mV (red), with an ionic strength of 1 mM, calculated without **a** and with **b** absorptive-dispersive interactions

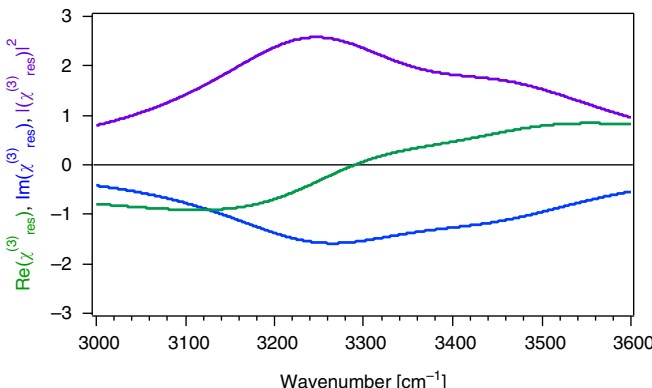

**Fig. 3** Resonant $\chi^{(3)}_{res}$ lineshape. Real and imaginary portions and square modulus of the $\chi^{(3)}_{res}$ response from a dual-oscillator system in the diffuse layer over a charged surface simulated for $\omega_1 = 3250$ cm$^{-1}$, $\gamma_1 = \gamma_2 = 0°$, $\omega_2 = 3450$ cm$^{-1}$, $\Gamma_1 = \Gamma_2 = 150$ cm$^{-1}$, $\chi^{(3)}_1 = 200$, and $\chi^{(3)}_2 = 100$

of OH oscillators, or solutions containing high concentrations of SHG or SFG active chromophores. Fresnel coefficients are not accounted for. In each example, we show the counterfactual of turning off absorptive-dispersive interactions to clearly demonstrate when they are important.

**Single oscillator.** In our analysis of equation (3) for the case of a single-frequency system (n = 1), we use $\omega_1 = 2100$ cm$^{-1}$, which is in the vicinity of nitrile or carbon monoxide systems reported in the literature[15–19]. With an arbitrary value of $A_1 M_1 = 10$, a damping factor, $\Gamma_v$, of 15 cm$^{-1}$, $\gamma_1 = 0$ degrees, and $\chi^{(3)} = 1.0$, we obtain SFG intensity maxima that move from 2095 cm$^{-1}$ at $\Phi(0)$ =-0.24 V to 2100 cm$^{-1}$ at zero mV to 2105 cm$^{-1}$ at + 0.24 V. While the lineshapes vary symmetrically with potential when absorptive-dispersive interactions are turned off (Fig. 1b), which is achieved by setting $\frac{\kappa}{\sqrt{\kappa^2+(\Delta k_z)^2}}e^{i\varphi}$ to 1 in equation (3c), the lineshapes are observed to vary asymmetrically with potential upon turning absorptive-dispersive interactions on (Fig. 1c). The latter case yields a shift of the peak frequency of $20 \pm 1$ cm$^{-1}$/V over this voltage range (Fig. 1d). Other input parameters yield different shifts; for instance, shifting $\varphi$ by 180° reverses the sign of the frequency shift. The shift and the lineshapes obtained here are reminiscent of what has been reported for dipole-dipole

coupling[20] and Stark shifts using SFG spectroscopy[15–19], even though the mechanism producing the simulated results described here invokes purely optical (absorptive and dispersive) interactions via equations (3) and (4).

**Two coupled oscillators.** Here, we take the example of a methyl symmetric ($\omega_1 = 2880$ cm$^{-1}$) and asymmetric ($\omega_1 = 2950$ cm$^{-1}$) stretching mode ($\gamma_1 = 0°$ and $\gamma_2 = 180°$) of a surface-bound molecule, perhaps one that is part of an organic field effect transistor. Using the values indicated in the caption, we obtain frequency shifts in the peak positions with potentials that are comparable to those found in the single-oscillator case. As shown in Fig. 2b, we find that the spectra shift symmetrically with the sign of the change in potential when absorptive-dispersive mixing is turned off, while turning it on leads to asymmetric lineshapes (Fig. 2b), even for the relatively modest potentials investigated here.

In Figs. 3 and 4, we present results that might be relevant for water in contact with a charged surface. We assume two OH stretching modes at 3200 and 3400 cm$^{-1}$ that are opposite in phase and that each have a 120 cm$^{-1}$ damping factor. Unlike in the prior two examples, we now include vibrational resonances in the $\chi^{(3)}$ term, according to 3c so as to account for the SFG response from the polarized water molecules in the diffuse layer. Equation (3c) shows that $\chi^{(3)}_{res}$ interacts with the static interfacial potential, $\Phi(0)$, via the $\chi^{(3)}$ phase angle $\varphi$, given by equation (4). Indeed, as established herein, the $\chi^{(3)}$ phase angle $\varphi$ is associated with $\Phi(0)$ as both depend on the Debye screening length. We omit the nonresonant $\chi^{(2)}_{NR}$ term for simplicity and emphasize that the $\chi^{(3)}_{res}$ term is not expressed as one would for, for instance, third harmonic generation. Instead, the resonance examined in our case of a water/solid or water/air interface is expressed in a similar manner as the $\chi^{(2)}_{res}$ term in equation (2) using two OH oscillators. We shift the OH resonance frequencies slightly to the blue (by $+50$ cm$^{-1}$) and give them slightly larger damping terms (150 cm$^{-1}$) when compared to $\chi^{(2)}_{res}$ so as to account for the presumably looser hydrogen-bonding strengths in the solvent molecules present in the diffuse layer when compared to those that interact with the interface. As shown in Fig. 3, the parameters chosen here produce real and imaginary parts of the $\chi^{(3)}_{res}$ response that are in reasonable agreement with the $\chi^{(3)}_B$ spectra reported by Wen et al[2]. yet, they are opposite in sign.

Figure 4 shows three cases and their counterfactuals. The first case recapitulates examples where the pH over an oxide surface is

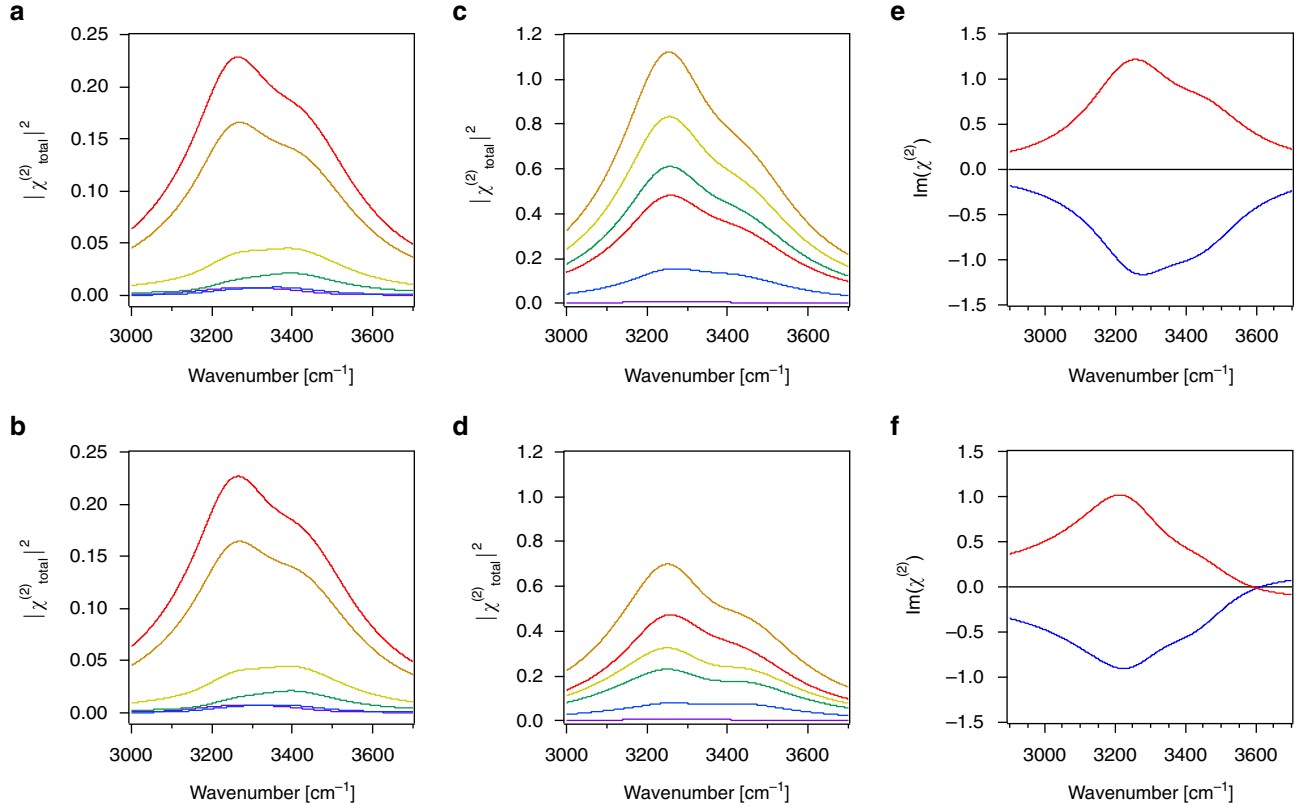

**Fig. 4** Charged solid/water interface. SFG intensity spectra for two oscillators on a charged surface simulated for $\omega_1 = 3200 \text{ cm}^{-1}$, $\gamma_1 = 0°$, $\omega_2 = 3400 \text{ cm}^{-1}$, $\gamma_2 = 180°$, $A_1M_1 = A_2M_2 = 10$, $\Gamma_1 = \Gamma_2 = 120 \text{ cm}^{-1}$, $\chi^{(3)}_1 = 500$, and $\chi^{(3)}_2 = 150$ using charge densities of + 0.001 (purple), −0.005 (blue), −0.025 (green), −0.05 (yellow), −0.17 (orange), and −0.25 (red) C/m$^2$ and a constant ionic strength of 0.1 M calculated without **a** and with **b** absorptive-dispersive interactions. **c**, **d** Same as in **a**, **b** but for ionic strengths that vary with charge density as in A and B as follows: 10 mM at + 0.001 C/m$^2$ (purple), 100 μM at −0.005 C/m$^2$ (blue), 40 μM at −0.025 C/m$^2$ (green), 40 μM at −0.05 C/m$^2$ (yellow), 100 μM at −0.17 C/m$^2$ (orange), and 10 mM −0.25 C/m$^2$ (red). **e**, **f** Im($\chi^{(2)}$) spectra with $\omega_1 = 3200 \text{ cm}^{-1}$, $\gamma_1 = 180°$, $\omega_2 = 3400 \text{ cm}^{-1}$, $\gamma_2 = 0°$, $A_1M_1 = A_2M_2 = 10$, $\Gamma_1 = \Gamma_2 = 120 \text{ cm}^{-1}$, $\chi^{(3)}_1 = 500$, and $\chi^{(3)}_2 = 250$ calculated using charge densities of + 0.2 (blue) and -0.2 (red) C/m$^2$ and a constant ionic strength of 100 μM, without **e** and with **f** absorptive-dispersive interactions

varied while maintaining a relatively high constant ionic strength, 100 mM in this example. In our example, charge densities for fused silica are estimated from work by Brown and co-workers[21]. Figures 4a, b show that the SFG intensity near 3200 cm$^{-1}$ increases with increasingly negative potential, which is consistent with recent reports by Borguet and co-workers[22]. As expected for short Debye lengths from equations (3) and (4) and Fig. 1a, absorptive-dispersive mixing is of minor importance for this case.

The next case is presented to recapitulate the experiment that is more commonly found in the literature, namely that of changing pH while not controlling for variations in ionic strength with pH. Ionic strengths in this case vary between tens of μM at circumneutral pH, given dissolved atmospheric CO$_2$, and 10 mM at pH 2 and 12. Figures 4c, d show that absorptive-dispersive interactions are considerable, changing even the order of the various simulated spectra shown in the two graphs, which is now understood from the fact that low ionic strengths, and thus long Debye lengths, correspond to relatively large $\chi^{(3)}$ phase angles $\varphi$. The responses discussed in this case are comparable to literature reports of metal/water[23] and fused silica/water[24–26] interfaces, yet they are obtained using the mixing of absorptive and dispersive terms that was not invoked in that prior work. This result indicates that unless the $\chi^{(3)}$ phase angle $\varphi$ is known, it is difficult to interpret the SFG spectra, be they heterodyned or not.

The final case shown in Fig. 4 examines charge densities that exceed those of common mineral/water interfaces. Here,

we consider the widely studied system of purely cationic and anionic lipid monolayers at the air/water interface, which normally carry plus or minus 0.1 to 0.2 C/m$^2$ surface charge[27]. The experiments are typically performed by adding small amounts of μM lipid solutions to ultrapure water contained in a Langmuir trough. In heterodyned, or phase-sensitive, SFG spectra, the sign of the Im($\chi^{(2)}$) is commonly taken to indicate an "up" or "down" orientation of the OH oscillators that are probed[28–30]. Indeed, Fig. 4e and f show positive and negative Im($\chi^{(2)}$) spectra for the hypothetical cases of a water surface covered with anionic and cationic lipids, respectively. Yet, turning on absorptive-dispersive mixing (Fig. 4f) is shown to lead to sign flips at certain frequencies (in the present case at 3600 cm$^{-1}$). This result is reminiscent of recent work by Yamaguchi and co-workers[31], but it is obtained without invoking changes in molecular orientation or phases for the various classes of oscillators contributing to the SFG response. We emphasize here that despite the use of only two oscillators in our model, the sign flip produced by the interactions discussed here could be easily mistaken for the presence of a third distinct oscillator. In an experiment, such an interpretation would be incorrect, emphasizing the necessity to account for the potential-dependent $\chi^{(3)}$ term.

**Three coupled oscillators**. Our final scenario examines the two broad oscillators studied in the previous section and a third oscillator at 3700 cm$^{-1}$ that we give a comparatively sharp

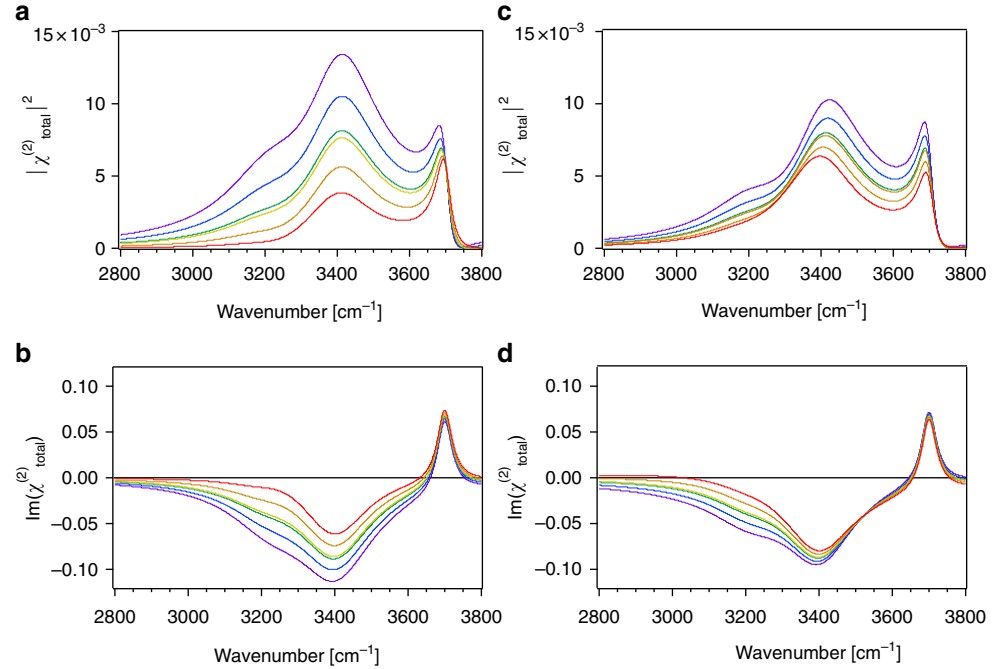

**Fig. 5** Air/water interface. Intensity and imaginary SFG spectra for three oscillators on a surface simulated for $\omega_1 = 3200$ cm$^{-1}$, $\gamma_1 = 0°$, $\omega_2 = 3400$ cm$^{-1}$, $\gamma_2 = 0°$, $A_1 M_1 = 2$, $A_2 M_2 = 10$, $\Gamma_1 = \Gamma_2 = 120$ cm$^{-1}$, $\omega_3 = 3700$ cm$^{-1}$, $\gamma_3 = 180°$, $A_3 M_3 = 2$, $\Gamma_3 = 25$ cm$^{-1}$, $\chi^{(3)}{}_1 = 200$, and $\chi^{(3)}{}_2 = 100$ calculated without **a**, **b** and with **c**, **d** absorptive-dispersive interactions for charge densities of + 0.0003, + 0.00015, + 0.000015, −0.0003, −0.00015, and −0.000015 C/m$^2$ and ionic strength of 40 μM

linewidth by providing it with a damping factor of 25 cm$^{-1}$. This system is set up to produce SFG responses that are reminiscent of those obtained from the air/water interface. Given reports that the surface of water may be acidic[32] or basic[33, 34], the presence of a non-zero potential at the air/water interface cannot be excluded, even though it is likely to be smaller than, for instance, the potential at a quartz/water interface held at pH 7 and low ionic strength. We therefore explore conditions that correspond to potentials of only up to 20 mV here.

Figure 5 shows that the 3200 cm$^{-1}$ mode in the intensity and Im($\chi^{(2)}$) spectra appears to shift towards lower wavenumbers and the intensity there increases as the charge density, and thus the potential, is raised to more negative values, whereas the sharp spectral feature at 3700 cm$^{-1}$ increases in intensity with a minor frequency shift. These intensity changes are comparable to those reported for some air/water[35–37] and ice/vapor[38] interfaces even though the mechanism giving rise to the SFG intensity changes described is purely based on absorptive-dispersive, i.e. optical, interactions. Yet, we also note that other combinations of amplitude, phase, and potential can produce different spectral features. Turning off absorptive-dispersive interactions (Figs. 5a, b) indicate substantial differences in the spectral lineshapes compared to the case where the interactions are turned on (Figs. 5c, d), indicating again that unless the $\chi^{(3)}$ phase angle $\varphi$ is known, it is difficult to interpret the SFG spectra of charged interfaces.

To further elaborate this point, Fig. 6 shows that the amplitude of the imaginary component of the SFG response in the 3000 to 3200 cm$^{-1}$ region of our simple three-oscillator system shown in Fig. 5 varies with interfacial potential. Comparing Fig. 6a and b, we find that absorptive-dispersive interactions result in a crossing point for the three curves that occurs at a potential that is considerably lower than what is found when absorptive-dispersive interactions are not taken into account. This result is of interest as this particular frequency region has been associated with some controversy regarding measured[28, 39–42] and

computed[28, 41, 43–45] SFG responses, highlighting the importance of properly determining all the phases of all contributing components in the nonlinear optical responses of charged interfaces. In particular, Fig. 6 seems to suggest that a zero Im ($\chi^{(2)}$) amplitude in the 3000 to 3200 cm$^{-1}$ frequency range indicates a net negative surface potential at the air/water interface.

## Discussion
Our mathematical modeling results show how vibrational SFG intensity spectra may be influenced by the presence of absorptive-dispersive interactions from an interfacial potential, $\Phi(0)$. Considerable spectral shifts with varying contributions of $\Phi(0)$ are observed in SFG intensity spectra simulated for single oscillator systems that approach in magnitude Stark shifts reported from SFG spectroscopy of charged interfaces, yet the model employed here is based purely on absorptive-dispersive interactions. For multi-oscillator systems, we find that spectral lineshapes vary substantially with applied potential and $\chi^{(3)}$ phase angle $\varphi$. Indeed, without invoking a microscopic interpretation other than that the potential decays from the surface at $z = 0$ to a value of zero at $z = \infty$, the model presented here produces, for some combinations of values of phase, amplitude, and potential, spectra that closely resemble pH- and ionic strength-dependent features in SFG intensity spectra that have been reported in the literature. Importantly, the agreement is good if one expresses the $\chi^{(3)}$ phase angle $\varphi$ as arctan($\Delta k_z / \kappa$). Such agreement with the measured SFG intensity spectra highlights the importance of accurately determining and analyzing the SFG spectral lineshapes, which has been heretofore mostly neglected in the literature.

Our results indicate that second-order electronic or vibrational resonances of charged interfaces are probably probed best by recording their real and imaginary components. Such an approach would unambiguously identify physics and chemistry separately from otherwise interfering optical interactions that may have absorptive-dispersive mixing as their origin. Heterodyning (HD) or phase-referencing (PR) are popular

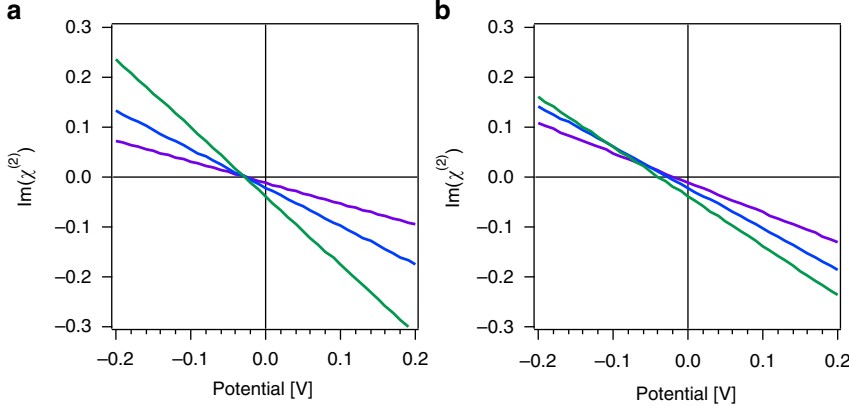

**Fig. 6** Potential-dependent offsets. Imaginary part of the SFG response at 3000 (purple), 3100 (blue), and 3200 (green) cm$^{-1}$ for three oscillators on a surface simulated using the parameters from Fig. 5 as a function of interfacial potential, $\phi(0)$, calculated without **a** and with **b** absorptive-dispersive interactions

methods for determining real and imaginary contributions in SHG[46] and SFG[30, 47–49] spectroscopy, but to determine the sign or phase for the potential-dependent phase angle $\varphi$, one would need to add a second heterodyning, or phase referencing, step. This second step would quantify the $\chi^{(3)}$ phase angle, $\varphi$, in equation (3c). This can only be realized when the various terms in the $\chi^{(2)}$ and the $\chi^{(3)}$ contributions can be well isolated from experimental data[2, 4, 5]. Yet, such an approach for determination of the phase angle $\varphi$ has not been realized for conditions of vibrational or electronic resonance. Doing so, however, opens the opportunity to quantify, for a given experimental geometry, i.e. $\Delta k_z$, the Debye length, $\kappa^{-1}$, in a model-independent fashion through equation (4), provided that surface potentials decay exponentially with distance, or through a different but analogous expression, provided a different distance dependence. For instance, for a linearly decaying potential of the form $\Phi(z) = a(z-b)$ for $0 \leq z \leq b$ and $\Phi(z) = 0$ for $z > b$, the $\chi^{(3)}$ term takes the form $\frac{a\chi^{(3)}}{i\Delta k_z}(e^{\Delta k_z b} - 1)$, with $ab = -\Phi(0)$ for $a < 0$ and $b > 0$. For cases where the electrostatic potential or charge screening in the surface layer is more complicated than the simple models discussed here, an experimental determination is necessary.

Until doubly heterodyned (DHD) or doubly phase-referenced (DPR) approaches are possible, the formalism presented here offers an opportunity to revisit published second-order spectra and to analyze them using equations (4) and (3c) and some approximations for the required parameters as discussed in this present work. We also recommend probing charged interfaces off electronic or vibrational resonances as the purely real terms produced by this method allow for more straightforward interpretation, as shown recently for the $\alpha$-quartz/water interface[4].

There is undoubtedly new physics and chemistry waiting to be discovered in the area of charged interfaces, and our claim is not to have the final word on this complicated topic. Instead, we present the formalism of absorptive-dispersive interactions and the new physical insight in equation (4), which expresses the $\chi^{(3)}$ phase angle $\varphi$ explicitly as a straight-forward function of experimental geometry and solution conditions from which Debye screening lengths can be determined experimentally. Conversely, given a known screening length, i.e. $1/\kappa$, and particular experimental geometries, i.e. $\Delta k_z$, one can directly compute the $\chi^{(3)}$ phase angle $\varphi$. Moreover, as $\Delta k_z$ is a function of the incident angles of the IR and visible beams, as well as a function of the upconverter and infrared frequencies, the resulting $\chi^{(3)}$ phase angle $\varphi$ and therefore the SFG lineshapes depend on these parameters as well.

Proper treatment of absorptive-dispersive interactions in simulated SFG spectra, such as those obtained from atomistic calculations[28, 41, 43–45, 50–52], is likely to be an important next step in this rapidly moving field, along with consideration of possibly absorptive properties of the potential dependent term. As such a treatment has not been considered in the vast literature on this subject, the existing interpretations of resonantly enhanced nonlinear optical responses from charged interfaces need to be carefully re-examined. We caution that the results presented here may also be critically important for multi-dimensional[53] or time-resolved[54] spectroscopic studies of interfaces, for geometries where the surface potential extends into directions other than the surface normal[55, 56], and for molecular systems in which non-Coulombic potentials, such as dipole potentials, are important[57, 58].

## Methods

**Simulations**. All simulated spectra displayed in the figures were generated using Wolfram Mathematica 11 and the equations included above. Mathematica Notebooks containing the code for each figure are provided in the Supplementary Information (see Supplementary Datas 1-7). The notebooks are set up such that varying the optical, solution, and oscillator parameters in the first portion of the notebooks allow for the straightforward generation of both Im($\chi^{(2)}$) and Abs($\chi^{(2)}$)$^2$ spectra with and without absorptive-dispersive interactions over a wide range of values.

As is common in the SFG community, the spectra are not corrected for the wavelength dependent Fresnel coefficients of the incoming and outgoing beams, though the $\Delta k_z$ is calculated on a wavelength-dependent basis. The refractive index of water at each point was approximated using the built-in Mathematica Interpolate function generated from experimental data from the literature[59–61].

**Data availability**. All relevant data are available from the authors upon request to the corresponding authors, with the notebooks used to generate the model spectra provided in the Supplementary Information.

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

## Acknowledgements

This work was supported by the US National Science Foundation (NSF) under its graduate fellowship research program (GRFP) award to P.E.O. We also acknowledge helpful discussions with Martin Thämer and Kramer Campen of the Fritz Haber Institute Berlin. H.-F.W. gratefully acknowledges support from the Shanghai Municipal Science and Technology Commission (Project No. 16DZ2270100) and Fudan University. F.M.G. gratefully acknowledges support from the NSF through award number CHE-1464916 and a Friedrich Wilhelm Bessel Prize from the Alexander von Humboldt Foundation.

## Author contributions

F.M.G. and H.-F.W. conceived of the idea. P.E.O., H.-F.W., and F.M.G. analyzed the data. The manuscript was written with substantial contributions from all authors.

## Additional information

**Competing interests:** The authors declare no competing financial interests.

**Change history:** A correction to this article has been published and is linked from the HTML version of this paper.

