## [Peer Review File · Nature Communications]

Reviewers' comments:

Reviewer #1 (Remarks to the Author):

In the manuscript titled as “Second-order spectral lineshapes from charged interface”, the authors discuss the impacts of the charge on the surface on the SFG spectra, with the calculation (rather than simulation). The information would be important for understanding the SFG spectra. In this respect, I found something interesting, and I believe that this could be useful for surface-specific spectroscopists. However, I am afraid that the scope of the paper is limited to the researchers in the community. One of the reasons why the authors fail to appeal the strength of their paper is that they do not connect their finding with the previous many studies on the charged interface. Indeed, I did not see so much novel finding on physics, which is important for broadening the audience of the authors’ paper, in my opinion. This lack of new physical insight may make the manuscript less attractive. Therefore, it is difficult for me to recommend the current manuscript for the publication of Nature Comm. with the current manuscript.

Detailed comments

1. To demonstrate the authors’s statement, evaluation of the individual terms for a realistic system beyond the model calculation is needed. In fact, so far, the theoreticians claimed that the MD simulation can successfully reproduces the experimentally measured SFG spectra without the $\chi^{(3)}$ contribution (see, for example, JCP 141, 18C502 (2014)). Although this potentially conflicts with the author’s statement, the authors could not mention anything about it. This would weaken the message of the paper.
2. Related with point 1, very important discussion about the bulk contribution on the water bending mode seems to be ignored. Tahara and Moirta insisted that the $\chi^{(3)}$ contribution governs that the signal, against Nagata and Bonn, and Skinner’s interpretation (I am quite suspicious whether the authors read Ref. 34 and the sequential papers citing Ref. 34 carefully.) Recently, Benderskii and co-worker support the interpretation of Bonn and Skinner by using positively and negatively charged surfactant. If the authors’s statement that “unless the $\chi^{(3)}$ phase angle is known, it is difficult to interpret the SFG data” is true, the current manuscript can provide some ‘cautions’ to the Benderskii’s paper and eventually may solve the conflict of these interpretations. Personally, I believe that the authors’ message does make sense, but since no discussion is presented, I cannot judge how the authors’ discussion affects these interpretations.
3. As such, although this manuscript potentially has strong impact on understanding the molecular structure near the charged interface, and may resolve many different interpretations of the SFG spectra proposed in the latest five years, it cannot say so much about the previous work. I do hope that by elaborating their calculation with these previous works much more carefully, the authors can make a manuscript with more impacts (but not current manuscript).
4. About the reference, I strongly recommend the authors to read the literatures –carefully- and take the reference in a well-balanced manner. For example, about the water-air interface, Morita and co-worker first simulated the SFG spectra in 2009. Bonn and co-workers first pointed out the absence of 3100 cm^{-1} peak at the water-air interface in 2015. These are clearly written, for example, in Ref. 35, whereas the authors seem not to be careful about these details.

Reviewer #2 (Remarks to the Author):

In this paper, the authors present a model calculation on the second-order spectral lineshape of a charged interface due to interplay of the X(2), X(3) and the interfacial potential. One to three coupled oscillator(s) are discussed as representative examples. They demonstrate that the spectral shape largely depends on the interfacial potential and the phase from the interplay of the coherent length and Debye length.

It is known that second-order optical technique is unique in probing buried charged interface. Very recently, phase relationships in SHG/SFG were derived and demonstrated in experiment, see Phys. Rev. Lett. 116, 016101(2016), J. Phys. Chem. C 120,9165 (2016), Nature communications 7, 13587 (2016). This work is based on these early developments. To my knowledge, the importance of the phase on spectral lineshape (spectral distortion of the bonded OH band) had been discussed theoretically and experimentally in the early work (Phys. Rev. Lett. 116, 016101(2016)). This paper didn't present deeper physical insight based on their model calculation, although it is technically helpful for accurate analysis of SHG/SFG spectrum, which makes this paper is important only for a specialized readership. Thus, I do not believe that the results and analysis in the paper are novel and general enough to interest a broad readership. Therefore, I do not recommend publication in Nature Communication.

Reviewer #3 (Remarks to the Author):

The manuscript NCOMMS-17-06611-T by Ohno et al reports theoretical investigations on SFG spectral lineshapes from charged interfaces. The authors used recently deduced formulism (Ref. 2-5 in the manuscript) and a coupled-oscillator model to simulate SFG spectra with respect to the interfacial potential, and address the significance of the observed spectral changes. While the reported work appears to be thorough, I have one major concern with respect to the innovativeness of the presented work, which is detailed below. Thus, I feel that it is only incremental advance that is presented, which does not warrant publication in Nature Communications but is rather appropriate for a more specialized journal. Details follow below: I am mainly concerned about new scientific insights of this work. The theory adopted in the author's calculation was deduced earlier and has been discussed by many groups from theoretical (Ref. 3, 5) and experimental (Ref. 2, 4) aspects. Gonella et. al. (Ref. 3) indicated how the absorptive-dispersive mixing leads to modification of second-order nonlinear optical response from charged interfaces through calculations. Wen et. al. (Ref. 2) used the same theoretical framework to analyze the experimentally deduced SFG spectra and showed that although absorptive-dispersive mixing causes different SFG lineshapes, the spectra can be analyzed with a self-consistent third-order susceptibility of liquid water. In addition, importance of the phase term ($iDkz$) in the spectral lineshape was emphasized in Ref. 2 and 5. The authors have to compare their findings with these existing literatures (including their previous reports) and address what new scientific insights one can learn from this work. Such discussion is missing in the introductory paragraphs and the main text. It is thus difficult to justify novelty and significance of the authors analysis.

Reviewer #1 (Remarks to the Author):

In the manuscript titled as “Second-order spectral lineshapes from charged interface”, the authors discuss the impacts of the charge on the surface on the SFG spectra, with the calculation (rather than simulation). The information would be important for understanding the SFG spectra. In this respect, I found something interesting, and I believe that this could be useful for surface-specific spectroscopists. However, I am afraid that the scope of the paper is limited to the researchers in the community. One of the reasons why the authors fail to appeal the strength of their paper is that they do not connect their finding with the previous many studies on the charged interface. Indeed, I did not see so much novel finding on physics, which is important for broadening the audience of the authors’ paper, in my opinion. This lack of new physical insight may make the manuscript less attractive. Therefore, it is difficult for me to recommend the current manuscript for the publication of Nature Comm. with the current manuscript.

We thank the reviewer for the comments that the “information would be important for understanding SFG spectra”, that the reviewer “found something interesting”, and that the reviewer believes “that this could be useful for surface-specific spectroscopists”. Yet, we disagree that the work is 1) limited to researchers in the community, 2) that we do not provide connections to previous studies of charged interfaces, and that the work 3) lacks new physical insight. Specifically, we now show a side-by-side comparison of our four cases in the form of results obtained using the case where absorptive-dispersive interactions are turned on, and its counterfactual, i.e. the case where absorptive-dispersive interactions are turned off. We believe that this approach of using counterfactuals elevates the main message of the manuscript without adding too much to its density and hope that this substantial modification will convince the reviewer.

We rationalize our response by addressing each of the reviewer’s detailed comments below.

Detailed comments

1. *To demonstrate the authors’s statement, evaluation of the individual terms for a realistic system beyond the model calculation is needed. In fact, so far, the theoreticians claimed that the MD simulation can successfully reproduces the experimentally measured SFG spectra without the $\chi(3)$ contribution (see, for example, JCP 141, 18C502 (2014)). Although this potentially conflicts with the author’s statement, the authors could not mention anything about it. This would weaken the message of the paper.*

We fully understand the reviewer’s concern. The interpretation of the nonlinear optical spectra from aqueous interfaces remains controversial. Yet, the new physical insight provided by Eqn. 4, which expresses the $\chi^{(3)}$ phase angle φ explicitly as a straight-forward function of experimental geometry and solution conditions will be invaluable not only to experimentalists but also to computational groups. With our newly identified formalism, counterfactuals can now finally be carried out, i.e. spectra can be computed with and without absorptive-dispersive interactions so that comparisons to experiments may be made more reliably.

We have added the following statement to page 2:

“Finally, our newly identified formalism provides a means to perform counterfactuals in computational studies of the nonlinear optical responses of charged interfaces, i.e. spectra can be now be computed with and without absorptive-dispersive interactions so that comparison to experiments can be made more reliably.”

To emphasize this point, we have included in Figures 1, 2, 4, and 5 the counterfactual of not including absorptive-dispersive interactions, with relevant new text marked in yellow to aid the reviewer.

On page 5, we now refer to Fig. 1A, which plots the $\chi^{(3)}$ phase angle that is given by eqn. 4 as a function of ionic strength, indicating its upper and lower bounds for an aqueous interface that is probed in the OH stretching region using an SFG experiment in a reflection geometry that employs incident angles of 45 degrees and 60 degrees for the upconverter and infrared frequencies, respectively, such that $\Delta k_z \approx 1/(44 \times 10^{-9} \text{ m})$.

On page 6, we now state

“In each example, we show the counterfactual of turning off absorptive-dispersive interactions to clearly demonstrate when they are important.”

Further down, on the same page, we now state

“While the lineshapes vary symmetrically with potential when absorptive-dispersive interactions are turned off, which is achieved by setting $\frac{\kappa}{\sqrt{\kappa^2 + (\Delta k_z)^2}} e^{i\varphi}$ to 1 in eqn. 3c (Fig. 1B), the lineshapes are observed to vary asymmetrically with potential upon turning absorptive-dispersive interactions on (Fig. 1C).”

On page 7, we state

“As shown in Figure 2A, we find that the spectra shift symmetrically with the sign of the change in potential when absorptive-dispersive mixing is turned off, while turning it on leads to asymmetric lineshapes (Fig. 2B), even for the relatively modest potentials investigated here.”

Page 8-11 contains new text outlining three cases of charged aqueous interfaces in the limits of low to moderate charge density (fused silica/water interface) at high and low ionic strength, and high charge density at low ionic strength (recapitulating lipid monolayer studies available in the literature). The text is marked for the reviewer’s convenience.

On page 11 we now state:

“There is undoubtedly new physics and chemistry waiting to be discovered in the area of charged interfaces, and our claim is not to have the final word on this complicated topic. Instead, we present the new formalism of absorptive-dispersive interactions and the new physical insight in Eqn. 4, which expresses the $\chi^{(3)}$ phase angle φ explicitly as a straight-forward function of experimental geometry and solution conditions from which Debye screening lengths can be determined experimentally. Conversely, given a known screening length, i.e. $1/\kappa$, and particular experimental geometries, i.e. Δk_z , one

can directly compute the $\chi^{(3)}$ phase angle φ . Moreover, as Δk_z is a function of the incident angles of the IR and VIS beams, as well as a function of the upconverter and infrared frequencies, the resulting $\chi^{(3)}$ phase angle φ and therefore the SFG lineshapes depend on these parameters as well.”

We also point out that the three attached graphs at the end of this response show experimental SFG intensity spectra from fused silica/water interfaces held at varying conditions of pH and ionic strength, and the analogous spectra computed using our formalism. The qualitative agreement is good, especially given the simplicity of our model. The quantitative agreement is of course subject to improvement once scientists take their geometric parameters (incident angles, Fresnel coefficients, etc) and experimental conditions of surface charge and ionic strength into account.

2. *Related with point 1, very important discussion about the bulk contribution on the water bending mode seems to be ignored. Tahara and Moirita insisted that the $\chi(3)$ contribution governs that the signal, against Nagata and Bonn, and Skinner’s interpretation (I am quite suspicious whether the authors read Ref. 34 and the sequential papers citing Ref. 34 carefully.) Recently, Benderskii and co-worker support the interpretation of Bonn and Skinner by using positively and negatively charged surfactant. If the authors’s statement that “unless the $\chi(3)$ phase angle is known, it is difficult to interpret the SFG data” is true, the current manuscript can provide some ‘cautions’ to the Benderskii’s paper and eventually may solve the conflict of these interpretations. Personally, I believe that the authors’ message does make sense, but since no discussion is presented, I cannot judge how the authors’ discussion affects these interpretations.*

We thank the reviewer for the comment that our “message makes sense” but we disagree that “no discussion is presented”. Our position is that whatever the current state of published spectral interpretations, including water’s quadrupole contributing (as put forth by Morita and co-workers), they need to be revisited in light of the newly identified interaction mechanism. We provide the formal framework of how to take this important step. The mention of “new chemistry and physics” in the new text on page 11, and the counterfactuals presented in the new Figures 1, 2, 4, and 5 speak to this effect quite clearly, encouraging other scientists to use the new formalism on their own data to refine their model interpretations.

We point out here that our position is not to take sides with one particular set of research groups, as the interpretation of the water surface spectra, be they from water stretching or bending modes, is still evolving. Instead, we offer an opportunity that is based on new physical insight for all to re-analyze their spectra using the newly identified absorptive-dispersive mixing mechanism with both the $\chi^{(2)}$ and the $\chi^{(3)}$ contributions.

3. *As such, although this manuscript potentially has strong impact on understanding the molecular structure near the charged interface, and may resolve many different interpretations of the SFG spectra proposed in the latest five years, it cannot say so much about the previous work. I do hope that by elaborating their calculation with these previous works much more*

carefully, the authors can make a manuscript with more impacts (but not current manuscript).

We very much appreciate the reviewer's statement that the work "potentially has strong impact on understanding the molecular structure near the charged interface, and may resolve many different interpretations of the SFG spectra proposed in the latest five years, ...". Again, we hope that the reviewer understands our position to not take sides, but to provide an impartial view of the current state of the field, while providing also new physical insight. We also hope that the reviewer will appreciate the side-by-side comparison we now provide in the revised Figures.

To further point out the importance of our new formalism to understanding the SFG spectra obtained from charged interfaces, we now provide two new Figures (4 E and F), in which we compare computed $\text{Im}(\chi^{(2)})$ spectra of water in contact with a positively and a negatively charged lipid monolayer. The analysis clearly shows $\text{Im}(\chi^{(2)})$ sign flips reminiscent of recent work by the Yamaguchi group once absorptive-dispersive mixing is turned on.

4. *About the reference, I strongly recommend the authors to read the literatures –carefully- and take the reference in a well-balanced manner. For example, about the water-air interface, Morita and co-worker first simulated the SFG spectra in 2009. Bonn and co-workers first pointed out the absence of 3100 cm^{-1} peak at the water-air interface in 2015. These are clearly written, for example, in Ref. 35, whereas the authors seem not to be careful about these details.*

We added these references to the main text and appreciate the fact that the reviewer asked us to include them. We caution that providing a comprehensive review of the SFG spectroscopy of aqueous surfaces is outside the scope of this work. Moreover, given that the discussion of the air/water interface represents only one of four cases presented in the manuscript, we hope that the reviewer understands our need to provide a balanced set of references, not just for the water spectra, but also for the other three systems we discuss.

Reviewer #2 (Remarks to the Author):

*In this paper, the authors present a model calculation on the second-order spectral lineshape of a charged interface due to interplay of the $X(2)$, $X(3)$ and the interfacial potential. One to three coupled oscillator(s) are discussed as representative examples. They demonstrate that the spectral shape largely depends on the interfacial potential and the phase from the interplay of the coherent length and Debye length. It is known that second-order optical technique is unique in probing buried charged interface. Very recently, phase relationships in SHG/SFG were derived and demonstrated in experiment, see *Phys. Rev. Lett.* 116, 016101(2016), *J. Phys. Chem. C* 120,9165 (2016), *Nature communications* 7, 13587 (2016). This work is based on these early developments. To my knowledge, the importance of the phase on spectral lineshape (spectral distortion of the bonded OH band) had been discussed theoretically and experimentally in the early work (*Phys. Rev. Lett.* 116, 016101(2016)). This paper didn't present deeper physical insight based on their model calculation, although it is technically helpful for accurate analysis of SHG/SFG spectrum, which makes this paper is important only for a specialized*

readership. Thus, I do not believe that the results and analysis in the paper are novel and general enough to interest a broad readership. Therefore, I do not recommend publication in Nature Communication.

We thank the reviewer for the comments and appreciate the opinion that the work is found to be “technically helpful”. Yet, we disagree that the work does not “provide deeper physical insight” and is therefore “important only for a specialized readership”.

Specifically, the text provided in the 2016 PRL paper cited by the reviewer does not explicitly mention absorptive-dispersive interactions, nor are outcomes on spectral lineshapes mentioned or discussed (even though spectra are certainly shown). Instead, the expressions described in that work use a “complex Psi” (eqn 2 and Figs S2 and S3 in the SI in that paper), with the units of IPsil given in mV. This formalism is difficult to follow, as static potentials are always real. In other words, time-invariant entities cannot interfere with one another. Our manuscript now states on page 3 the following:

“Even though during the past two decades SFG studies of charged aqueous interfaces have essentially neglected the $\chi^{(3)}$ contributions, recent work by Tian and Shen and coworkers (ref. 2 in the main text) firmly established its existence and attempted to separate the $\chi^{(2)}$ and $\chi^{(3)}$ contributions to the total SFG signal from the air/water interface of electrolyte aqueous solution, making the discussion presented here ever more relevant. The $\chi^{(3)}$ phase angle φ is not the phase of the “complex Ψ ” discussed in reference (2). Instead, the phase matching factor we provide is for the optical field, not the static field, given that static potentials are real. As the $\chi^{(3)}$ phase angle φ is determined by the coefficients of the real and imaginary $\chi^{(3)}$ terms of eqn. 3b, it ultimately depends on the optical parameters and electrostatic conditions of the experiment.”

This formalism leads directly to eqn. 4, which, in turn, directly leads to the new physical insight that is now described in the statement on top of page 4: “Eqn. 4 reveals that the $\chi^{(3)}$ phase angle φ contains rich new information, as it depends on the values of κ and Δk_z that are given by each specific condition of the sample, aqueous environment, pH, ionic strength, temperature, etc., and the geometry of the laser experiment. Indeed, given that the $\chi^{(3)}$ phase angle φ is in principle measurable, as indicated below, and given that Δk_z is known for a given experiment, eqn. 4 is a means for experimentally finding the Debye screening length as a function of charge density, ionic strength, and surface potential. As such, eqn. 4 provides an experimental benchmark for evaluating both common mean field theories as well as atomistic models of charged interfaces.”

Therefore, our position is that eqn. 4 will be of considerable use to scientists in the field as it is *the* tool for including absorptive-dispersive mixing that is needed for the re-interpretation of published data.

Reviewer #3 (Remarks to the Author):

The manuscript NCOMMS-17-06611-T by Ohno et al reports theoretical investigations on SFG spectral lineshapes from charged interfaces. The authors used recently deduced formalism (Ref. 2-5 in the manuscript) and a coupled-oscillator

model to simulate SFG spectra with respect to the interfacial potential, and address the significance of the observed spectral changes. While the reported work appears to be thorough, I have one major concern with respect to the innovativeness of the presented work, which is detailed below. Thus, I feel that it is only incremental advance that is presented, which does not warrant publication in Nature Communications but is rather appropriate for a more specialized journal.

We thank the reviewer for the comments and appreciate the statement that “the reported work appears to be thorough”. Yet, we respectfully disagree that the work lacks innovation.

Details follow below:

I am mainly concerned about new scientific insights of this work. The theory adopted in the author’s calculation was deduced earlier and has been discussed by many groups from theoretical (Ref. 3, 5) and experimental (Ref. 2, 4) aspects.

We disagree that the work has been discussed by many groups. Only Shen and co-workers, Roke and co-workers, and we have published on this new topic. The cited prior work does not mention or discuss absorptive-dispersive interactions beyond saying, as in our own work, refs (4-5) that “the terms may mix”. The area is rapidly evolving, warranting, in our opinion, the chosen submission format. We have changed the manuscript substantially from its original version, focusing specifically on the presentation and discussion of counterfactuals for the various cases we have discussed. The goal is to indicate the clear difference in outcomes obtained when including or excluding the mechanism of absorptive-dispersive interactions.

Gonella et. al. (Ref. 3) indicated how the absorptive-dispersive mixing leads to modification of second-order nonlinear optical response from charged interfaces through calculations.

While Reviewer 3 states that “Gonella et. al. (Ref. 3) indicated how absorptive-dispersive mixing leads to modification of second-order nonlinear optical response from charged interfaces through calculations”, this outcomes on SFG spectral lineshapes is not explicitly mentioned or discussed. Instead, a statement in Ref. 3 regarding on-resonance conditions is limited to two clauses in the second-to-last paragraph, which states “The presented description is only relevant insofar as the main aqueous phase is probed, i.e., for vibrational SFG experiments that center on the O–H stretch or bending mode (e.g., reviewed in ref 57), resonant SHG experiments that focus on the charge transfer to solvent mode insofar the water is also resonantly excited (e.g., ref 58), and nonresonant SHG measurements that probe the response of all noncentrosymmetric molecules in the sample (e.g., refs 11 and 59–61).”

The main focus of this reference is on the intensity of the observed signals using non-resonant second harmonic generation (SHG); upon reading the paper and without themselves performing the sort of calculations discussed in our manuscript, a reader is unlikely to be fully aware of the dramatic change in lineshapes themselves that may also occur. This is what motivates our belief that a discussion such as ours is warranted, and new.

Wen et. al. (Ref. 2) used the same theoretical framework to analyze the experimentally deduced SFG spectra and showed that although absorptive-dispersive mixing causes different SFG lineshapes, the spectra can be analyzed with a self-consistent third-order susceptibility of liquid water.

The expressions described in that work use a “complex Psi” (eqn 2 and Figs S2 and S3 in the SI in that paper), with the units of lPsi given in mV. This formalism is difficult to follow, as static potentials are always real. In other words, time-invariant entities cannot interfere with one another. We have modified our manuscript to now state on page 3 the following:

“Even though during the past two decades SFG studies of charged aqueous interfaces have essentially neglected the $\chi^{(3)}$ contributions, recent work by Tian and Shen and coworkers (ref. 2 in the main text) firmly established its existence and attempted to separate the $\chi^{(2)}$ and $\chi^{(3)}$ contributions to the total SFG signal from the air/water interface of electrolyte aqueous solution, making the discussion presented here ever more relevant. The $\chi^{(3)}$ phase angle φ is not the phase of the “complex Ψ ” discussed in reference (2). Instead, the phase matching factor we provide is for the optical field, not the static field, given that static potentials are real. As the $\chi^{(3)}$ phase angle φ is determined by the coefficients of the real and imaginary $\chi^{(3)}$ terms of eqn. 3b, it ultimately depends on the optical parameters and electrostatic conditions of the experiment.”

Moreover, eqn. 3b leads directly to the new physical insight that is now described on page 4:

“Eqn. 4 reveals that the $\chi^{(3)}$ phase angle φ contains rich new information, as it depends on the values of κ and Δk_z that are given by each specific condition of the sample, aqueous environment, pH, ionic strength, temperature, etc., and the geometry of the laser experiment. Indeed, given that the $\chi^{(3)}$ phase angle φ is in principle measurable, as indicated below, and given that Δk_z is known for a given experiment, eqn. 4 is a means for experimentally finding the Debye screening length as a function of charge density, ionic strength, and surface potential. As such, eqn. 4 provides an experimental benchmark for evaluating both common mean field theories as well as atomistic models of charged interfaces.”

Therefore, our position is that eqn. 4 will be of considerable use to scientists in the field as it is *the* tool for including absorptive-dispersive mixing that is needed for the re-interpretation of published data.

In addition, importance of the phase term (iDk_z) in the spectral lineshape was emphasized in Ref. 2 and 5.

We fully agree with the reviewer that the phase term $i\Delta k_z$ is not new. Our manuscript makes proper reference to the work by Wen et al. and Gonella et al., who first reported on its importance, and to our prior work, which provided direct experimental proof for its existence. The present work gives the form of the $\chi^{(3)}$ phase angle φ , as shown in eqn. 4.

The authors have to compare their findings with these existing literatures (including their previous reports) ...

While we understand the reviewer’s request, we respectfully disagree that a comparison to the existing literature is missing. It is beyond the scope of this manuscript to comprehensively cite the entirety of work previously published in this field, and we believe that the prior art is properly cited in the introduction and

throughout the four examples we provide in the manuscript, with an eye towards a fair balance. It is also clearly indicated in our manuscript that one common element missing in the literature is the concept, and indeed the physical picture, of the absorptive-dispersive interactions that our work establishes to be necessary for understanding the SHG and SFG spectroscopy of charged surfaces and interfaces.

... and address what new scientific insights one can learn from this work. Such discussion is missing in the introductory paragraphs and the main text. It is thus difficult to justify novelty and significance of the authors analysis.

We appreciate that the reviewer is requesting clear statements regarding what new physical insight our work provides. Our formalism states that the $\chi^{(3)}$ phase angle φ is associated with the optical field, which is in fact complex. Indeed, the phase matching factor we provide is for the optical field, not the static field (as written in ref. 2), providing the new physical insight summarized in eqns 3 and 4. Our work is the first to explicitly derive this formalism, the relevant equations, and give several concrete examples of interest to the broader community.

To summarize, we emphasize these points in the main text as follows:

Page 2:

We derive that the $\chi^{(3)}$ phase angle φ contains important physical parameters pertaining to the experimental geometry and the Debye screening length, thus opening a path to testing existing electrical double layer theories against an experimental measurable.

Finally, our newly identified formalism provides a means to perform counterfactuals in computational studies of the nonlinear optical responses of charged interfaces, i.e. spectra can now be computed with and without absorptive-dispersive interactions so that comparison to experiments can be made more reliably.

Page 3:

“Even though during the past two decades SFG studies of charged aqueous interfaces have essentially neglected the $\chi^{(3)}$ contributions, recent work by Tian and Shen and coworkers (ref. 2 in the main text) firmly established its existence and attempted to separate the $\chi^{(2)}$ and $\chi^{(3)}$ contributions to the total SFG signal from the air/water interface of electrolyte aqueous solution, making the discussion presented here ever more relevant. The $\chi^{(3)}$ phase angle φ is not the phase of the “complex Ψ ” discussed in reference (2). Instead, the phase matching factor we provide is for the optical field, not the static field, given that static potentials are real. As the $\chi^{(3)}$ phase angle φ is determined by the coefficients of the real and imaginary $\chi^{(3)}$ terms of eqn. 3b, it ultimately depends on the optical parameters and electrostatic conditions of the experiment.”

Page 4:

Eqn. 4 reveals that the $\chi^{(3)}$ phase angle φ contains rich new information, as it depends on the values of κ and Δk_z that are given by each specific condition of the sample, aqueous environment, pH, ionic strength, temperature, etc., and the geometry of the laser experiment. Indeed, given that the $\chi^{(3)}$ phase angle φ is in principle measurable, as indicated below, and given that Δk_z is known for a given experiment,

eqn. 4 is a means for experimentally finding the Debye screening length as a function of charge density, ionic strength, and surface potential. As such, eqn. 4 provides an experimental benchmark for evaluating both common mean field theories as well as atomistic models of charged interfaces.

Page 6

In each example, we show the counterfactual of turning off absorptive-dispersive interactions to clearly demonstrate when they are important.

and

While the lineshapes vary symmetrically with potential when absorptive-dispersive interactions are turned off, which is achieved by setting $\frac{\kappa}{\sqrt{\kappa^2 + (\Delta k_z)^2}} e^{i\varphi}$ to 1 in eqn. 3c (Fig. 1B), the lineshapes are observed to vary asymmetrically with potential upon turning absorptive-dispersive interactions on (Fig. 1C).

Page 7

As shown in Figure 2A, we find that the spectra shift symmetrically with the sign of the change in potential when absorptive-dispersive mixing is turned off, while turning it on leads to asymmetric lineshapes (Fig. 2B), even for the relatively modest potentials investigated here.

Page 8-11 contains new text outlining three cases of charged aqueous interfaces in the limits of low to moderate charge density (fused silica/water interface) at high and low ionic strength, and high charge density at low ionic strength (recapitulating lipid monolayer studies available in the literature). The text is marked for the reviewers convenience.

Page 11

There is undoubtedly new physics and chemistry waiting to be discovered in the area of charged interfaces, and our claim is not to have the final word on this complicated topic. Instead, we present the new formalism of absorptive-dispersive interactions and the new physical insight in Eqn. 4, which expresses the $\chi^{(3)}$ phase angle φ explicitly as a straight-forward function of experimental geometry and solution conditions from which Debye screening lengths can be determined experimentally. Conversely, given a known screening length, i.e. $1/\kappa$, and particular experimental geometries, i.e. Δk_z , one can directly compute the $\chi^{(3)}$ phase angle φ . Moreover, as Δk_z is a function of the incident angles of the IR and VIS beams, as well as a function of the upconverter and infrared frequencies, the resulting $\chi^{(3)}$ phase angle φ and therefore the SFG lineshapes depend on these parameters as well.

Page 12

As such a treatment has not been considered in the vast literature on this subject, the existing interpretations of resonantly enhanced nonlinear optical responses from charged interfaces need to be carefully reexamined.

REVIEWERS' COMMENTS:

Reviewer #1 (Remarks to the Author):

The authors have revised the manuscript greatly. I find the newly added figures are very useful and appealing. Furthermore, I find very good motivation of the authors to revise the manuscript. As such, I am much more positive about this manuscript than used to be. Furthermore, I have no special request to revise the manuscript.

Reviewer #2 (Remarks to the Author):

After reading through the response and the revised manuscript, I think my major concern about the manuscript has been partially clarified. In the response, the authors state that "...Instead, the expressions described in that work use a "complex Psi" (eqn 2 and Figs S2 and S3 in the SI in that paper), with the units of $|\Psi|$ given in mV. " It is well-known that the surface potential and $\chi(3)$ should not depend on measurement. Mathematically, one may separate the term of phase mismatch from ϕ and $\chi(3)$, or attach it to either one, but the physics and the math are equivalent to that established in the literature. I don't think the authors from early literature didn't understand this point. Nevertheless, I do agree the discussion on the spectral lineshape in this work is important for SFG/SHG community when analyzing a charged interface. Therefore, I do not object to publication of this work in Nature Comm.

Reviewer #3 (Remarks to the Author):

The manuscript was revised to have detailed descriptions about the author's achievement and its relation to the existing works. I agree that the formalism presented in the manuscript has a new form in terms of $\chi(3)$ phase angle, which may make the absorptive-dispersive mixing effect easily understandable. I also agree with the authors that the absorptive-dispersive mixing effect is discussed in the manuscript very explicitly and carefully. However, the formalism presented in the manuscript is still under the same theoretical framework, as these developed in Ref. 2-5, and, thus, cannot provide new physical insight. In addition, the absorptive-dispersive mixing effect has been indicated in Ref. 2-5. Therefore, I think that the presented work is only incremental advance but lacks distinguishable originality and far-reaching insight. I do not recommend publication in Nature Communication.